# Impact of the COVID-19 Pandemic on the Clinical Profile of Candidemia and the Incidence of Fungemia Due to Fluconazole-Resistant *Candida parapsilosis*

**DOI:** 10.3390/jof8050451

**Published:** 2022-04-27

**Authors:** Antonio Ramos-Martínez, Ilduara Pintos-Pascual, Jesús Guinea, Andrea Gutiérrez-Villanueva, Edith Gutiérrez-Abreu, Judith Díaz-García, Ángel Asensio, Reyes Iranzo, Isabel Sánchez-Romero, María Muñoz-Algarra, Víctor Moreno-Torres, Jorge Calderón-Parra, Elena Múñez, Ana Fernández-Cruz

**Affiliations:** 1Unidad de Enfermedades Infecciosas, Servicio de Medicina Interna, Hospital Universitario Puerta de Hierro, 28222 Majadahonda, Spain; jorge050390@gmail.com (J.C.-P.); elena.munez@salud.madrid.org (E.M.); anafcruz999@gmail.com (A.F.-C.); 2Instituto de Investigación Sanitaria Puerta de Hierro—Segovia de Arana (IDIPHSA), 28222 Majadahonda, Spain; 3Servicio de Medicina Interna, Hospital Universitario Puerta de Hierro, 28222 Majadahonda, Spain; ilduarapintos@gmail.com (I.P.-P.); a.gutierrezv@hotmail.com (A.G.-V.); edithgutierrez3010@gmail.com (E.G.-A.); victor.moreno.torres.1988@gmail.com (V.M.-T.); 4Clinical Microbiology and Infectious Diseases, Hospital General Universitario Gregorio Marañón, 28007 Madrid, Spain; jguineaortega@yahoo.es (J.G.); juddithdiaz5@gmail.com (J.D.-G.); 5Instituto de Investigación Sanitaria Gregorio Marañón, 28009 Madrid, Spain; 6CIBER Enfermedades Respiratorias-CIBERES (CB06/06/0058), 28029 Madrid, Spain; 7Servicio de Medicina Preventiva, Hospital Universitario Puerta de Hierro, 28222 Majadahonda, Spain; aasensiov@salud.madrid.org; 8Servicio de Anestesia y Reanimación, Hospital Universitario Puerta de Hierro, 28222 Majadahonda, Spain; mariareyes.iranzo@salud.madrid.org; 9Servicio de Microbiología, Hospital Universitario Puerta de Hierro, 28222 Majadahonda, Spain; msromero@salud.madrid.org (I.S.-R.); algarra18@hotmail.com (M.M.-A.)

**Keywords:** COVID-19, candidemia, *Candida parapsilosis*, drug resistance, microbial, fluconazole

## Abstract

Severely ill COVID-19 patients are at high risk of nosocomial infections. The aim of the study was to describe the characteristics of candidemia during the pre-pandemic period (January 2019–February 2020) compared to the pandemic period (March 2020–September 2021). Antifungal susceptibilities were assessed using the EUCAST E.Def 7.3.2 broth dilution method. Fluconazole-resistant *C. parapsilosis* isolates (FRCP) were studied for sequencing of the *ERG11* gene. The incidence of candidemia and *C. parapsilosis* bloodstream infection increased significantly in the pandemic period (*p* = 0.021). ICU admission, mechanical ventilation, parenteral nutrition and corticosteroids administration were more frequent in patients with candidemia who had been admitted due to COVID-19. Fifteen cases of FRCP fungemia were detected. The first case was recorded 10 months before the pandemic in a patient transferred from another hospital. The incidence of FRCP in patients admitted for COVID-19 was 1.34 and 0.16 in all other patients (*p* < 0.001). ICU admission, previous *Candida* spp. colonization, arterial catheter use, parenteral nutrition and renal function replacement therapy were more frequent in patients with candidemia due to FRCP. All FRCP isolates showed the Y132F mutation. In conclusion, the incidence of candidemia experienced an increase during the COVID-19 pandemic and FRCP fungemia was more frequent in patients admitted due to COVID-19.

## 1. Introduction

The COVID-19 pandemic has represented a substantial burden on the health care activity of hospitals [1]. Admission to medical wards and ICUs, together with the frequent invasive procedures to which these patients are subjected, could facilitate the occurrence of nosocomial infections and the emergence of antimicrobial-resistant infections [2,3].

The increase in fungal infections has been one of the major concerns in the management of patients with COVID-19 [4,5]. Although these patients do not usually present some typical risk factors for candidemia, such as neutropenia or abdominal surgery, several studies have shown a high incidence of candidemia. This fact has been related to treatment with steroids or immunosuppressants, ICU admission and mechanical ventilation [6,7]

*Candida parapsilosis* ranges between the second and third most frequent species causing invasive yeast infections [8]. This species is characterized by a tendency to form biofilms on medical devices and colonize the hands of healthcare personnel, which may contribute to invasive infections and nosocomial outbreaks [9,10]. Although azoles (especially fluconazole and voriconazole) are the treatment of choice for invasive infections due to *C. parapsilosis*, recent studies noted an increase in the number of hospital outbreaks due to azole-resistant *C. parapsilosis* [10,11]. The emergence of those outbreaks may be associated with previous use of fluconazole and/or antibiotics, or the clonal spread of isolates across the hospital [9].

We recently observed an increase in the number of cases of candidemia alongside the beginning of COVID-19 pandemic in our institution. Some of these bloodstream infections were produced by fluconazole-resistant *Candida parapsilosis* (FRCP). This study describes the clinical profile of candidemia before and during the COVID-19 pandemic and the incidence of fungemia due to fluconazole-resistant *Candida parapsilosis*.

## 2. Methods

### 2.1. Setting, Patients and Study Design

This retrospective, observational, single-institution study was conducted in a 620-bed tertiary university hospital located in Madrid, Spain. Between January 2019 and September 2021, blood cultures positive for *Candida* spp. were studied. The onset of the COVID-19 pandemic outbreak in March 2020 in Spain led to consider two distinct periods in the study. A pre-pandemic period (period 1), comprising January 2019–February 2020, and a pandemic period (period 2), from March 2020 to September 2021.

We retrospectively reviewed the electronic medical records of all adult patients (aged 18 years or older) with candidemia. An episode of candidemia was defined as the detection of at least one blood culture positive for *Candida* species. In addition to demographic data, we collected medical history, underlying diseases, Charlson index [12], treatments prior to the episode of candidemia, admission to the ICU, and the most important risk factors for candidemia, including previous antibiotherapy, catheters, parenteral nutrition, renal replacement techniques and previous surgery. The treatment of the candidemia episode was also recorded. Given the observational nature of the research, patients were managed according to routine clinical care.

### 2.2. Definitions

In the case of a patient presenting more than one episode of candidemia, only the first one was considered. Obesity was defined as a body mass index greater than 30 kg/m^2^. Among the category of immunosuppressive treatment, any treatment with immunosuppressive agents administered prior to admission for COVID-19 due to chronic disease or transplantation was considered.

### 2.3. Microbiological Studies

The patient was considered to have COVID-19 if they presented consistent symptoms (fever, cough and radiological infiltrate) together with a positive reverse transcriptase polymerase chain reaction (RT-PCR) result performed by one of the diagnostic systems available in the hospital [13,14].

#### 2.3.1. Blood Cultures and Antifungal Susceptibility Testing

Blood cultures were obtained by standard procedures and processed with the BD BACTEC FX (Becton Dickinson, Sparks, MD, USA). All systems were applied according to the manufacturer’s instructions. When the blood culture was positive and the Gram stain demonstrated the presence of a yeast, a subculture was performed in BBL CHROMagar Candida Medium (Becton Dickinson TM). The yeasts were identified by MALDI-TOF MS (Bruker Daltonic TM).

#### 2.3.2. Antifungal Susceptibility Testing and *ERG11* Gene Sequencing

In vitro antifungal susceptibilities to amphotericin B, fluconazole, voriconazole, and posaconazole (Sigma-Aldrich, Madrid, Spain) of isolates were assessed by the EUCAST EDef 7.3.2 broth dilution method. Isolates were categorized as resistant and/or non-wild type according to EUCAST clinical breakpoints. Fluconazole-resistant *C.*
*parapsilosis* isolates were further studied for *ERG11* gene sequencing, as previously reported [15].

#### 2.3.3. Microsatellite Typing Procedure

Species-specific microsatellite markers were used to genotype all *C.*
*parapsilosis* isolates (CP1, CP4a, CP6 and B), as previously reported [16].

### 2.4. Data Analysis

Quantitative variables were reported as median and interquartile range (IQR), and categorical variables as counts (%). The chi-square test or Fisher exact test were used to compare the distribution of categorical variables, and Student’s t-test or Mann–Whitney U test were used for quantitative variables. Significance was set at a *p* value of less than 0.05.

### 2.5. Ethical Statement

The study was approved by the Institutional Review Board (CEIm) at Hospital Universitario Puerta de Hierro-Majadahonda, and a waiver for the informed consent was granted (PI_154_2020). The study complied with the provisions in EU and Spanish legislation on data protection and the Declaration of Helsinki 2013.

## 3. Results

During the study period, we observed 88 episodes of candidemia in 88 patients: 29 episodes in period 1 (January 2019–February 2020) and 56 episodes in period 2 (March 2020–September 2021). The incidence rate per 10,000 patient days during period 1 was 1.36 (0.93–1.93) and 2.55 (2.01–3.19) during period 2 (*p* = 0.002). The incidence in this second period was 3.18 (1.82–4.89; *p* = 0.006) in patients admitted for COVID-19, 13 and 2.43 (1.85–2.43) in patients without COVID-19 (*p* = 0.006).

The 91 isolates of *Candida* species from the 88 positive blood cultures were distributed as follows: *C. albicans* (*n* = 38; 41.8%), *C. parapsilosis* (*n* = 31 isolates; 34.1%), *C. glabrata* (*n* = 15; 16.5%), *C. tropicalis* (*n* = 3; 3.3%) and *C. krusei* (*n* = 3; 3.3%) (Figure 1).

During period 1, 31 isolates were identified in 29 patients and distributed as *C. albicans* (*n* = 13; 34.2%), *C. parapsilosis* (*n* = 6; 19.4%) and *C. glabrata* (*n* = 5; 16.1%). During period 2, 60 *Candida* isolates were identified in 59 patients and distributed as *C. albicans* (*n* = 25; 41.7%), *C. parapsilosis* (*n* = 24; 40%) and *C. glabrata* (*n* = 7; 11.7%). Incidence of *C. parapsilosis* bloodstream infection significantly increased in period 2 (*p* = 0.024). However, the comparison of the incidence of blood tract infection by *C. albicans* and *C. glabrata* between the two periods showed no significant differences (*p* = 0.431 and *p* = 0.276, respectively).

When comparing patients without COVID from both periods (before and after the onset of the pandemic), no differences were detected in the most relevant variables. Differences reaching statistical significance were not found in any of the variables studied: median age, 70 years (IQR 59–76 years) vs. 65 years (59–75 years; *p* = 0.653); solid tumor (34.5% vs. 25.6%; *p* = 0.290); chronic renal disease (10.3% vs. 7%; *p* = 0.462); Charlson comorbidity index [2 (IQR 1–4) vs. 3 (IQR 1–5); *p* = 0.251)]; hospital stay before candidemia [20 days (IQR 9–36 days) vs. 20 days (IQR 7–37 days); *p* = 0.595]; ICU admission (10.3% vs. 16.3%; *p* = 0.561); central venous catheter as source of candidemia (42.3% vs. 46.5%; *p* = 0.443); recurrent candidemia (4% vs. 7.5%; *p* = 0.500); mortality (58.6% vs. 55.8%; *p* = 0.504). Likewise, there were no significant differences regarding the species causing the bloodstream infections: *C. albicans* (44.8% vs. 46.5%; *p* = 0.44.6%); *C. parapsilosis* (24.1% vs. 34.9%; *p* = 0.174); *C. glabrata* (17.2% vs. 16.3%; *p* = 0.456).

### 3.1. Candidemia Episodes in Patients Hospitalized for COVID-19

The clinical characteristics of patients with candidemia in patients admitted for COVID-19 compared to non-COVID-19 patients are shown in Table 1.

Patients with COVID-19 tended to be older (*p* = 0.077), more frequently had obesity and had a lower incidence of neoplastic diseases (Table 1). Certain risk factors for candidemia, such as catheter use and parenteral nutrition, were more frequent in patients admitted for COVID-19, while previous abdominal surgery was less frequent (Table 1). The use of a central venous catheter as the source of candidemia was more frequent in COVID-19 patients. A total of 15 patients with COVID-19 (93.7%) were admitted to the ICU when candidemia was detected (all required mechanical ventilation) vs. only 11 patients (13.3%) without COVID-19 (*p* < 0.001); additionally, 15 out of 16 patients admitted for COVID-19 (93.7%) received corticosteroids prior to the onset of candidemia (*p* < 0.001). There was 1 patient admitted due to COVID-19 (6.25%) and 8 additional non-COVID-19 patients (11.1%) who did not receive antifungal treatment for the episode of candidemia because it was considered a futile measure due to the extreme worsening of the patient. The hospital mortality rate was high in both groups (43.8% in COVID-19 and 58.3% in non-COVID-19 patients, respectively; *p* = 0.216). Distribution of *Candida* species causing candidemia according COVID-19 status at admission is presented in Table 2.

### 3.2. Candidemia Episodes Due to Fluconazole-Resistant Candida parapsilosis

Regarding resistance to antifungal agents, 20 isolates were resistant to fluconazole (22%); of those, 15 isolates corresponded to FRCP, 3 isolates were *C. krusei*, (intrinsically resistant to fluconazole), 1 isolate *C. glabrata* and 1 isolate *C. blankii*, respectively. One isolate of *C. glabrata* was resistant to posaconazole. There were no cases of resistance to liposomal amphotericin B. Of the 15 FRCP isolates, 10 (66.7%) were also resistant to voriconazole. There were two FRCP isolates concurrently resistant to caspofungin.

FRCP was detected on 2 blood cultures (6.7%) during period 1 compared with 13 in period 2 (21.7%, *p* = 0.054). The first patient with FRCP candidemia was detected in May 2019 in a patient with a ventricular assist system transferred from a hospital in a different region (Extremadura) to be assessed for cardiac transplantation. FRCP candidemia was detected in 14 additional patients throughout the study period (Figure 1). The incidence rate per 10,000 patient days during period 1 was 0.0937, while it was 0.389 during period 2 (*p* = 0.032). The incidence of FRCP in patients admitted due to COVID-19 was 1.34, whereas it was 0.16 in the rest of the patients (*p* < 0.001).

The clinical characteristics of patients with FRCP bloodstream infection as compared to the remaining patients are shown in Table 3. A total of 9 episodes of candidemia due to FRCP (53.3%) were detected in the ICU compared with 16 episodes produced by other *Candida* strains (15.3%, *p* < 0.001). Candidemia due to FRCP was more frequent in the surgical ICU than in the medical ICU and was associated with previous *Candida* spp. colonization, arterial catheter use, parenteral nutrition and renal function replacement therapy (Table 3). Candidemia due to FRCP was not associated with higher hospital mortality. The characteristics of patients with fungemia caused by FRCP compared to those produced by fluconazole-susceptible *C. parapsilosis* are illustrated in Table 4.

The 15 fluconazole-resistant *C. parapsilosis* isolates were subjected to sequencing of the *ERG11* gene and the A395T mutation, conferring the amino acid substitution Y132F was found in all of them. Such mutation was never found in fluconazole-susceptible isolates studied. All isolates harboring the A395T resulted to be a single genotype that demonstrated that all of them belonged to the same clone.

## 4. Discussion

Our investigation has shown an increase in the incidence of candidemia during the pandemic period, mainly due to an increase in *C. parapsilosis* cases. The most remarkable finding was the increase FRCP bloodstream infection episodes in patients admitted due to COVID-19, which, to our knowledge, has not been reported to date.

### 4.1. Candidemia Episodes in Patients Hospitalized for COVID-19

We detected a higher incidence of candidemia during the pandemic period, and in particular, it was higher in patients admitted for COVID-19, as previously described [6], suggesting that specific risk factors may be involved in this complication.

Classic risk factors for candidemia, such as cancer or previous surgery (abdominal or of other site), were less common among patients with COVID-19, for instance, all patients who had an underlying hematologic disease were in the group of patients without COIVD-19. Similar to our findings, other authors report fewer cases of patients with neoplasms in patients with COVID-19 [7]. Not surprisingly, obesity, diabetes and chronic lung disease tended to be present more frequently in patients with candidemia and COVID-19, as they are known COVID-19 risk factors that increase the probability hospital admission [5,17]. The limited number of cases in our series may have precluded detection of the possible association of candidemia in patients with COVID-19 with chronic lung disease or diabetes.

A distinctive feature of patients with COVID-19 who developed candidemia is the association with admission to the ICU [5,6,7,17]. As expected, patients admitted to a conventional ward had a much lower incidence due to the absence of risk factors for candidemia. Patients with severe COVID-19 who required admission to the ICU presented a high incidence of antibiotic use, corticosteroid treatment, central venous catheterization, parenteral nutrition and mechanical ventilation. Reducing the duration of admission to the ICU and the risk factors present in each patient as much as possible is an elusive goal, but one that could decrease this high incidence.

The common use of corticosteroids (which is a known risk factor for candidemia) in COVID-19 patients has been involved in their immune impairment and could explain in part the increase in candidemia observed in similar investigations [17,18]. This association was to be expected considering the previously described correlation with episodes of candidemia in oncology patients as well as in patients with other underlying diseases [19].

Regarding the species causing the bloodstream infection, other studies have observed a prominent role of non-albicans *Candida* species in patients admitted for COVID-19 [20,21]. The present study underlines the relevant role of *C. parapsilosis* in these patients that has not been previously reported [5,6,17,22]. *C. parapsilosis* commonly colonizes the skin, so that any manipulation or alteration of the skin integrity—such as those that occur frequently during ICU admission, e.g., catheter use—would facilitate candidemia and explain the high percentage of cases in this setting. Many of these candidemia episodes were originated in catheters.

### 4.2. Candidemia Episodes Due to Fluconazole-Resistant Candida parapsilosis

The most relevant finding of our study was the increase in cases of candidemia caused by FRCP during the pandemic. In this study, we found the Y132F mutation in *ERG11* gene only in fluconazole-resistant isolates and not in fluconazole-susceptible isolates. This study constitutes the second series of cases of FRCP due to the Y132F mutation reported in Spain [23]. Although all 15 isolates were resistant to fluconazole, susceptibility to other azoles varied. Most of them were also resistant to voriconazole. Previous use of fluconazole has been considered a risk factor for the development of resistance in several *Candida* species; however, no statistically significant difference was detected in our patients (Table 3 and Table 4) [24,25]. We hypothesize that the patient’s proximity to the environmental reservoir may have played a more important role than previous antifungal exposure. In relation to the other resistant species, it is worth highlighting the case of bloodstream infection by *C. blankii*, which was detected before the pandemic. This species could become an emerging epidemiological problem due to its reduced sensitivity to azoles and echinocandins [26].

When comparing patients with candidemia due to FRCP both to those with fluconazole-susceptible *C. parapsilosis*, and with non-parapsilosis candidemia, the former presented more surgical ICU admissions, arterial catheters, a renal function replacement technique and previous *Candida* colonization (any species). Compared with non-parapsilosis candidemia, parenteral nutrition was also associated with FRCP candidemia. Since multivariate analysis was not performed, it is not possible to establish which variables are the more commonly associated with this infection. However, other studies suggest that vascular access and surgical ICU admission are the most relevant parameters [27]. It should be taken into account that epidemiological surveillance cultures are not usually performed in conventional hospital wards, but, on the contrary, they are performed in ICUs. This may account for the high proportion of previous *Candida* colonization detected in patients with FRCP fungemia. Other studies with a high number of azole-resistant *Candida* infections have observed a clear association with prolonged hospital admission [28]. In the present study, there was a non-significant trend to a longer hospital stay in patients with FRCP.

The COVID-19 pandemic may have provided an optimal scenario for the spread of a resistant pathogen that already existed in the hospital, as a result of transferring a first patient with candidemia with FRCP from another hospital. Undoubtedly, the high number of patients admitted to the ICU has supposed a somewhat unusual clinical scenario and it cannot be ruled out that the usual aseptic measures to control the horizontal transmission of pathogens were reduced. It is noteworthy that the *Candida* species involved in this outbreak is characterized by its ability to adhere to catheters and to colonize the hands of healthcare personnel [9,10,24]. We also are unaware of whether there were any particularly contaminated areas that could have acted as a reservoir and were not adequately managed. The fact that these outbreaks tend to affect one of the ICUs more severely than other ICUs in the same institution suggests the existence of an environmental reservoir that is often very difficult to determine [24,29,30].

There are some similarities between candidemia outbreaks due to *C. auris* described to date in patients with COVID-19 and those by FRPC observed in our series, such as the identification of patients with infection or colonization by the same *Candida* species during the preceding months [31,32]. Other relevant characteristics of *C. auris* cases are the predominant involvement of certain areas of the ICUs, their relationship with prolonged hospital stays, previous administration of antifungal drugs and the high frequency of previous colonization by *Candida* spp. [31,32,33]. Additionally, its association with central venous catheters and the suspicion that the relaxation of infection control measures may have favored horizontal transmission is noteworthy [31]. These facts make it advisable to pursue early diagnosis and treatment of these infections in patients with COVID-19 admitted to the ICU and to intensify measures to control hospital-acquired infections.

In contrast to what has been observed in other FRCP outbreaks, candidemia due to this microorganism was not associated with increased mortality [10,24,29]. Our interpretation is that the Y132F substitution confers azole resistance but not increased virulence. It cannot be ruled out that the lower comorbidity in our patients coinfected with COVID-19 compared to other patients with candidemia may have influenced their prognosis.

Our findings should be a warning about the risk of transmission of resistant microorganisms among COVID-19 patients and the inform the appropriate preventive measures that need to be undertaken.

### 4.3. Limitations

Our study has several limitations. The most important limitation is the small number of patients with candidemia due to FRCP, which prevented the performance of a multivariate analysis to identify the variables most closely related to this infection. Since this is a single-center study, our conclusions may not be applicable to other institutions. The retrospective collection of patient information from the medical records could have prevented us from correctly recording some variables, such as obesity.

## Figures and Tables

**Figure 1 jof-08-00451-f001:**
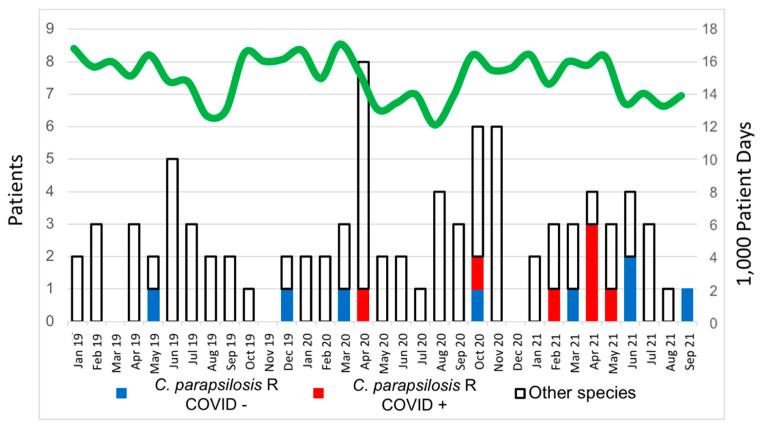
Incidence of candidemia from January 2019 to September 2021. R—resistant.

**Table 1 jof-08-00451-t001:** Clinical characteristics of patients with candidemia according to hospital admission due to COVID-19 or not.

	Patients with COVID-19 (*n* = 16)	Patents without COVID-19 (*n* = 72)	*p*-Value
Age (years), (median (IQR))	73.5 (66.5–77.5)	66 (59–76.5)	0.077
Male gender	12 (75)	49 (68.1)	0.413
Obesity	4 (25)	2 (2.8)	0.009
Diabetes mellitus	7 (43.8)	20 (27.8)	0.169
Heart failure	2 (12.5)	9 (12.5)	0.682
Chronic lung disease	5 (31.3)	9 (12.5)	0.076
Dementia	0	7 (9.7)	0.252
Chronic liver disease	1 (6.3)	9 (12.5)	0.270
Chronic renal failure	1 (6.3)	6 (8.3)	0.626
Solid tumor	1 (6.3)	21 (29.2)	0.047
Hematologic malignancy	0	6 (8.3)	0.288
Charlson comorbidity index, (median (IQR))	1 (0–3)	2 (1–4)	0.110
Solid organ transplantation	0	6 (8.3)	0.288
Hospital stay before candidemia, (median (IQR))	22.5 (14–53.5)	20 (4–35)	0.915
ICU admission	15 (93.7)	10 (13.9)	<0.001
Previous corticosteroids treatment	14 (87.5)	10 (13.9)	<0.001
Previous tocilizumab treatment	14 (87.5)	10 (13.9)	<0.001
Previous immunosuppressive treatment	0	6 (8.3)	0.288
Previous antifungal treatment	5 (31.3)	19 (26.4)	0.454
Previous antibiotic treatment	16 (100)	62 (86.1)	0.119
Central venous catheter	16 (100)	53 (73.6)	0.013
Arterial catheter	8 (50)	16 (22.2)	0.018
Parenteral nutrition	12 (75)	30 (41.7)	0.016
Renal replacement therapy	4 (26.7)	16 (21.9)	0.459
*Candida* spp. previous colonization	5 (31.3)	15 (20.8)	0.276
Candida Score	2 (1–2.5)	2 (0–3)	0.885
Abdominal surgery	1 (6.3)	21 (29.2)	0.047
Surgery, other site	0	14 (19.4)	0.046
Candidemia source			
Venous catheter	13 (81.3)	34 (47.2)	0.013
Intraabdominal	1 (6.3)	15 (20.8)	0.156
Urinary tract	1 (6.3)	8 (11.1)	0.316
Unknown	3 (18.7)	22 (30.6)	0.184
Fever	15 (93.8)	65 (90.3)	0.553
Endophtalmitis ^1^	3 (18.7)	5 (7.1)	0.094
Endocarditis ^2^	1 (6.3)	5 (6.9)	0.701
Recurrent candidemia	4 (26.7)	3 (6.2)	0.019
Acute kidney failure	4 (25)	35 (48.6)	0.073
Mortality	7 (43.8)	42 (58.3)	0.216
Mortality attributable to candidemia	1 (6.25)	6 (8.3)	0.660

IQR—interquartile range. ^1^ Fundoscopy was performed in 61 patients. ^2^ Echocardiography was performed in 73 patients.

**Table 2 jof-08-00451-t002:** *Candida* species causing candidemia in patients with and without COVID-19.

	Patients with COVID-19 (*n* = 16)	Patients without COVID-19 (*n* = 75) ^1^	*p*-Value
*C. albicans*	6 (37.5)	32 (42.7)	0.315
*C. parapsilosis*	9 (56.2)	22 (29.3) ^2^	0.019
FR *C. parapsilosis*	7 (43.7)	8 (10.7)	0.003
*C. glabrata*	1 (6.2)	12 (16)	0.175
*C. tropicalis*	0	3 (4)	0.280
*C. krusei*	0	3 (4)	0.280
*C. lusitaniae*	0	1 (1.3)	0.412
*C. africana*	0	1 (1.3)	0.412
*C. blankii*	0	1 (1.3)	0.412
Mixed ^3^	0	3 (4)	0.280

^1^ A total of 75 *Candida* species were isolated from 72 patients. FR—fluconazole resistant. ^2^ One case with blood cultures growing *Candida orthopsilosis* was included in the *C. parapsilosis* group. ^3^ Mixed: *C. albicans* plus *C. africana*, *C. albicans* plus *C. parapsilosis* and *C. parapsilosis* plus *C. krusei* (one patient each).

**Table 3 jof-08-00451-t003:** Clinical characteristics of patients with candidemia caused by fluconazole-resistant *C. parapsilosis* (FRCP) compared to non-FRCP.

	Fluconazole-Resistant *C. parapsilosis* (*n* = 15)	Other Strains (*n* = 73)	*p*-Value
Age, (years) (median (IQR))	67 (50–75)	69 (61–77)	0.063
Male gender	13 (86.7)	48 (65.8)	0.094
Obesity	1 (6.7)	5 (6.8)	0.730
Diabetes mellitus	5 (33.3)	22 (30.1)	0.514
Heart failure	1 (6.7)	10 (13.7)	0.402
Chronic lung disease	4 (26.7)	10 (13.7)	0.189
Dementia	0	7 (9.7)	0.252
Chronic liver disease	1 (6.7)	5 (6.8)	0.478
Chronic renal failure	1 (6.7)	6 (8.2)	0.659
Solid tumor	2 (13.3)	22 (27.4)	0.211
Hematologic malignancy	0	6 (8.2)	0.314
Charlson (median (IQR))	1 (0–2)	2 (1–4)	0.134
Solid organ transplantation	2 (13.3)	4 (5.5)	0.270
Hospital stay (median (IQR))	31 (17–51)	20 (5–36)	0.985
ICU admission	9 (60)	16 (21.9)	0.005
ICU stay ^1^ (median (IQR))	41.5 (6–60.5)	14 (4.5–27)	0.139
Medical ICU admission	2 (13.3)	9 (12.3)	0.438
Surgical ICU admission	7 (46.7)	7 (9.6)	<0.001
Previous corticosteroids treatment	8 (53.3)	16 (22.2)	0.019
Previous immunosuppressive treatment	2 (13.3)	4 (5.5)	0.270
Previous antibacterial treatment	14 (93.3)	64 (87.7)	0.460
Previous antifungal treatment	7 (46.7)	17 (23.3)	0.066
Previous azole treatment	4 (26.7)	8 (11)	0.118
Previous fluconazole treatment	3 (20)	6 (8.2)	0.178
Previous echinocandin treatment	4 (26.7)	9 (12.3)	0.152
Previous liposomal amphotericin B treatment	1 (6.7)	3 (4.1)	0.533
Central venous catheter	14 (93.3)	55 (75.3)	0.110
Arterial catheter	7 (77.8)	17 (29.8)	0.009
Parenteral nutrition	12 (80)	30 (41.1)	0.006
Renal replacement therapy	8 (53.3)	12 (16.4)	0.004
*Candida* spp. previous colonization	7 (46.7)	13 (17.8)	0.023
Candida Score	2 (1–3)	2 (0–3)	0.499
Abdominal surgery	2 (13.3)	20 (27.4)	0.211
Surgery, other site	4 (26.7)	10 (13.7)	0.189
COVID-19	7 (46.7)	8 (11)	<0.001
Fever	14 (93.3)	66 (90.4)	0.589
Endophtalmitis ^2^	1 (6.7)	7 (10)	0.570
Endocarditis ^3^	0	6 (8.2)	0.314
Recurrent candidemia	4 (26.7)	3 (4.1)	0.015
Acute kidney failure	8 (53.39	31 (42.5)	0.312
Candidemia source			
Venous catheter	10 (66.7)	37 (50.7)	0.200
Intraabdominal	3 (20)	13 (17.8)	0.545
Urinary tract	0	9 (12.3)	0.170
Unknown	2 (13.3)	14 (19.2)	0.455
Treatment ^4^			
Azole	9 (60)	32 (43.8)	0.195
Liposomal amphotericin B treatment	7 (46.7)	16 (21.9)	0.055
Echinocandin treatment	11 (73.3)	26 (35.6)	0.028
Mortality	7 (46.7)	42 (57.5)	0.312
Mortality attributable to candidemia	2 (22.2)	5 (8.9)	0.247

IQR—interquartile range. ^1^ ICU stay until the onset of candidemia. ^2^ Fundoscopy was performed in 61 patients. ^3^ Echocardiography was performed in 73 patients. ^4^ Some patients received more than one type of antifungal drug.

**Table 4 jof-08-00451-t004:** Clinical characteristics of patients with bloodstream infection due to *Candida parapsilosis* according to resistance to fluconazole.

	Fluconazole-Resistant *C. parapsilosis* (*n* = 15)	Fluconazole-Susceptible *C. parapsilosis* (*n* = 16)	*p*-Value
Age (years) (median (IQR))	67 (50–75)	70.5 (57–75)	0.309
Male gender	13 (86.7)	14 (87.5)	0.675
Obesity	1 (6.7)	0	0.484
Diabetes mellitus	5 (33.3)	5 (31.3)	0.602
Heart failure	1 (6.7)	3 (18.8)	0.325
Chronic lung disease	4 (26.7)	2 (12.5)	0.295
Chronic liver disease	1 (6.7)	0	0.484
Chronic renal failure	1 (6.7)	2 (12.5)	0.525
Solid tumor	2 (13.3)	5 (31.3)	0.224
Hematologic malignancy	0	2 (12.5)	0.258
Neutropenia	1 (6.7)	3 (18.8)	0.325
Charlson (median (IQR))	1 (0–2)	3 (2–3)	0.256
Solid organ transplantation	2 (13.3)	0	0.226
Hospital (stay median (IQR))	31 (17–51)	21 (3–30)	0.489
ICU admission	9 (60)	3 (18.8)	0.023
Medical ICU admission	2 (13.3)	1 (6.3)	0.475
Surgical ICU admission	7 (46.7)	2 (12.5)	0.043
Previous corticosteroids treatment	8 (53.3)	3 (18.8)	0.050
Previous immunosuppressive treatment	2 (13.3)	1 (6.3)	0.475
Previous antibacterial treatment	14 (93.3)	15 (93.8)	0.742
Previous antifungal treatment	7 (46.7)	6 (37.5)	0.439
Previous azole treatment	4 (26.7)	2 (12.5)	0.295
Previous fluconazole treatment	3 (20)	1 (6.3)	0.275
Previous echinocandin treatment	4 (26.7)	4 (25)	0.618
Previous liposomal amphotericin B treatment	1 (6.7)	0	0.484
Central venous catheter	14 (93.3)	15 (93.8)	0.742
Arterial catheter	7 (77.8)	3 (23.1)	0.017
Parenteral nutrition	12 (80)	10 (62.5)	0.250
Renal replacement therapy	8 (53.3)	2 (12.5)	0.019
*Candida* spp. previous colonization	7 (46.7)	2 (12.5)	0.044
Candida Score	2 (1–3)	2 (0–2)	0.413
Abdominal surgery	2 (13.3)	4 (25)	0.359
Surgery, other site	4 (26.7)	3 (18.8)	0.461
COVID-19	7 (46.7)	2 (12.5)	0.044
Candidemia source			
Venous catheter	10 (66.7)	10 (62.5)	0.553
Intraabdominal	3 (20)	3 (18.8)	0.642
Urinary tract	0	2 (12.5)	0.258
Unknown	2 (13.3)	1 (6.3)	0.475
Fever	14 (93.3)	15 (93.8)	0.742
Endophtalmitis ^1^	1 (6.7)	1 (6.3)	0.742
Endocarditis ^2^	0	1 (6.3)	0.516
Recurrent candidemia	4 (26.7)	0	0.043
Acute kidney failure	8 (53.3)	5 (31.3)	0.189
Mortality	7 (46.7)	7 (43.8)	0.578
Mortality attributable to candidemia	2 (13.3)	0	0.226

IQR—interquartile range. ^1^ Fundoscopy was performed in 21 patients. ^2^ Echocardiography was performed in 27 patients.

## Data Availability

The data supporting the reported results are available and can be obtained upon request by email to the corresponding author.

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
