# Peer review of "Impact of the COVID-19 Pandemic on the Clinical Profile of Candidemia and the Incidence of Fungemia Due to Fluconazole-Resistant Candida parapsilosis"

_jof, 2022, doi:10.3390/jof8050451_

Round 1

Reviewer 1 Report

It is a well-developed manuscript, the objectives are clear and the methodology is consistent, which contributes to the current knowledge of the events in relation to the emergence of infections of resistant microorganisms during the current COVID 19 pandemic, congratulations to the authors.

Author Response

We sincerely appreciate the considerations made by reviewer 1 about this work.

Reviewer 2 Report

In the article "Impact of COVID-19 pandemic on the clinical profile of candidemia and the incidence of fungemia due to fluconazole-resistant Candida parapsilosis" the authors evaluated the incidence of candidemia in a hospital during two periods, one before and one during the covid epidemic. They mainly discuss the occurrence of fluconazole-resistant Candida parapsilosis.

This reviewer requests the following adjustments/explanations:

  1. Correct the spelling 30kg/m2, on page 2, line 81, to 30kg/m2.
  2. In Figure 1, as the number of patients per month is low, and the average patient/day ratio is not measured, it would be better if a curve of the average patient/day for each month were superimposed; for example, the month of April/20 had the highest frequency of Candida infections. But what was the number of patients in this month? It could have been much higher than the other months. You can't tell by looking at the figure. The descriptions in the figure (legends) could be better written and made consistent with the text (initial month capitalized, C. parapsilosis instead of C. parap).
  3. On page 4, lines 126 to 131, information could be included about which Candida species associations were found. The information is in Table 2, but in another context.
  4. Does the information “No differences in the most relevant variables were detected between patients without COVID-19 diagnosed in period 1 with respect to period 2” refer to only patients without Covid-19 in both periods, or in the second period to all patients, with and without Covid-19? Confirm information.
  5. Check Table 1, Chronic liver disease information does not contain %, and parenteral nutrition is without open parenthesis. Candidemia source and Mortality informations need to be separated. It is a little confusing. This also occurs in other tables.
  6. (…) 3 isolates were C. kruseii (page 6, line 173). Correct krusei, and answer me, does C. krusei not show intrinsic resistance to fluconazole? Wouldn't this information be important to remember in the text? And about C. blankii, is this resistance common?
  7. Correct the scientific names throughout the text by spelling them in italics, including in figures and tables.
  8. Page 9, lines 251 to 253, the authors say "In this study, we found the Y132F mutation in ERG11 only in fluconazole-resistant isolates and not in susceptible isolates from the same episode”. What “the same episode” the authors refer to? It was unclear. Did resistant and susceptible isolates occur simultaneously in the same patient from the same blood sample?
  9. Line 290, page 10, delete "?”.

Author Response

Reviewer 2:
In the article "Impact of COVID-19 pandemic on the clinical profile of candidemia and the incidence of fungemia due to fluconazole-resistant Candida parapsilosis" the authors evaluated the incidence of candidemia in a hospital during two periods, one before and one during the covid epidemic. They mainly discuss the occurrence of fluconazole-resistant Candida parapsilosis.

This reviewer requests the following adjustments/explanations:

  1. Correct the spelling 30kg/m2, on page 2, line 81, to 30kg/m2.

This error has been corrected

  1. In Figure 1, as the number of patients per month is low, and the average patient/day ratio is not measured, it would be better if a curve of the average patient/day for each month were superimposed; for example, the month of April/20 had the highest frequency of Candida infections. But what was the number of patients in this month? It could have been much higher than the other months. You can't tell by looking at the figure. The descriptions in the figure (legends) could be better written and made consistent with the text (initial month capitalized, C. parapsilosis instead of C. parap).

A line showing the number of patients/day for each month has been added to this figure. The figure descriptions (legends) have also been corrected as suggested by the reviewer for greater uniformity.

  1. On page 4, lines 126 to 131, information could be included about which Candida species associations were found. The information is in Table 2, but in another context.

We agree with the referee's suggestion. We have incorporated a sentence on the statistical significance of the comparison between the two periods of the incidence bloodstream infection due to C. albicans and C. glabrata

  1. Does the information “No differences in the most relevant variables were detected between patients without COVID-19 diagnosed in period 1 with respect to period 2” refer to only patients without Covid-19 in both periods, or in the second period to all patients, with and without Covid-19? Confirm information.

Clearly, it was necessary to make it more precise that the patients diagnosed in the second period referred to in this paragraph were patients who did not present Covid. The associated sentence has been changed to give it a more accurate meaning

  1. Check Table 1, Chronic liver disease information does not contain %, and parenteral nutrition is without open parenthesis. Candidemia source and Mortality informations need to be separated. It is a little confusing. This also occurs in other tables.

We have corrected the errors in the parentheses and moved the information on the origin source of candidemia (Tables 1 and 4) to a different place (higher in the table).

  1. (…) 3 isolates were C. kruseii (page 6, line 173). Correct krusei, and answer me, does C. krusei not show intrinsic resistance to fluconazole? Wouldn't this information be important to remember in the text? And about C. blankii, is this resistance common?

We appreciate the referee's comments. The spelling of krusei has been corrected. We have added that all C. krusei were intrinsically resistant to fluconazole and have also added information in the discussion about the commonly reduced antifungal sensitivity that characterizes C. blankii and the risk of being a future epidemiologic problem.

  1. Correct the scientific names throughout the text by spelling them in italics, including in figures and tables.

We have traced the entire manuscript in an attempt to detect and correct such errors.

  1. Page 9, lines 251 to 253, the authors say "In this study, we found the Y132F mutation in ERG11 only in fluconazole-resistant isolates and not in susceptible isolates from the same episode”. What “the same episode” the authors refer to? It was unclear. Did resistant and susceptible isolates occur simultaneously in the same patient from the same blood sample?

In this study, we found the Y132F mutation in ERG11 only in fluconazole-resistant isolates and not in fluconazole-susceptible isolates.

There was an error in the sentence. We have clarified that the mutation was not present in fluconazole-susceptible isolates.

  1. Line 290, page 10, delete "?”.

The wrong question mark has been removed.